# Mapping quantal touch using 7 Tesla functional magnetic resonance imaging and single-unit intraneural microstimulation

Rosa Maria Sanchez Panchuelo[1*†], Rochelle Ackerley[2,3†], Paul M Glover[1], Richard W Bowtell[1], Johan Wessberg[2], Susan T Francis[1‡], Francis McGlone[4‡§]

[1]Sir Peter Mansfield Imaging Centre, School of Physics and Astronomy, University of Nottingham, Nottingham, United Kingdom; [2]Department of Physiology, University of Gothenburg, Göteborg, Sweden; [3]Laboratoire de Neurosciences Intégratives et Adaptatives, Aix-Marseille University, Marseille, France; [4]School of Natural Sciences and Psychology, Liverpool John Moores University, Liverpool, United Kingdom

**Abstract** Using ultra-high field 7 Tesla (7T) functional magnetic resonance imaging (fMRI), we map the cortical and perceptual responses elicited by intraneural microstimulation (INMS) of single mechanoreceptive afferent units in the median nerve, in humans. Activations are compared to those produced by applying vibrotactile stimulation to the unit's receptive field, and unit-type perceptual reports are analyzed. We show that INMS and vibrotactile stimulation engage overlapping areas within the topographically appropriate digit representation in the primary somatosensory cortex. Additional brain regions in bilateral secondary somatosensory cortex, premotor cortex, primary motor cortex, insula and posterior parietal cortex, as well as in contralateral prefrontal cortex are also shown to be activated in response to INMS. The combination of INMS and 7T fMRI opens up an unprecedented opportunity to bridge the gap between first-order mechanoreceptive afferent input codes and their spatial, dynamic and perceptual representations in human cortex.

*For correspondence: rosa.panchuelo@nottingham.ac.uk

†These authors contributed equally to this work
‡These authors also contributed equally to this work

Present address: §Institute of Psychology, Health and Society, University of Liverpool, Liverpool, United Kingdom

Competing interests: The authors declare that no competing interests exist.

## Introduction

The primary somatosensory cortex (S1) has been extensively explored in animal studies where it has been shown that this area displays multiple, fine-grained representations of the body (*Paul et al., 1972*; *Kaas et al., 1979*; *Favorov et al., 1987*). Penfield and Boldrey (*Penfield & Boldrey 1937*) derived the first maps of the somatotopic human body representation in S1 using electrical stimulation of the cortical surface. Somatosensory research in humans has involved using psychophysical (*Klatzky et al., 1985*; *Gescheider et al., 2002*), microneurographic (*Vallbo & Johansson 1984*; *Johansson & Vallbo 1983*), and neuroimaging (*McGlone et al., 2002*; *Martuzzi et al., 2014*; *Servos et al., 2001*) techniques to study different stages and levels of detail in somatosensory function. Functional magnetic resonance imaging (fMRI) has been used extensively for the non-invasive study of the somatosensory cortices in humans (*Nelson and Chen, 2008*; *McGlone et al., 2002*; *Sanchez-Panchuelo et al., 2010*). Most fMRI studies have investigated the spatial pattern of cortical activation in response to vibrotactile (*Francis et al., 2000*; *Sanchez-Panchuelo et al., 2010*) or pneumatic (*Huang and Sereno, 2007*; *Overduin and Servos, 2008*) mechanical stimulation of the digits, or to electrical stimulation of the skin (*Blankenburg et al., 2003*) or median nerve (*Kampe et al.,*

**eLife digest** The skin contains multiple types of sensory nerves that inform the brain about events occurring on the surface of the body. One way to study how this process works is to insert a very fine needle through the skin to stimulate a single sensory nerve with a small electrical current. This technique – known as intraneural microstimulation – can activate touch responses in the brain without an object actually contacting the skin.

Another technique called functional magnetic resonance imaging (fMRI) has been used to measure brain activity. These studies have revealed that when objects come into contact with the skin of the fingers, they stimulate several sensory nerves at the same time, which results in brain activity in a region called the somatosensory cortex.

Sanchez Panchuelo, Ackerley et al. combined fMRI and intraneural microstimulation to map brain activity in response to the activation of individual sensory nerves in the fingers of human volunteers. The experiments show that intraneural stimulation activates many areas of the brain that are also activated by mechanical contact. Future work will use this new method to study the brain's response to signals from different types of sensory nerves.

*2000*; *Ferretti et al., 2007*). These approaches excite large populations of different classes of mechanoreceptive afferents resulting in relatively diffuse activations in contralateral S1 and bilateral secondary somatosensory cortex (S2).

Microneurography provides a method to record the spike discharge activity of a single mechanoreceptive afferent in conscious humans (*Vallbo and Hagbarth, 1968*) to determine its response to skin contact and the properties of its receptive field, i.e. location, size, and shape. In this manner, mechanoreceptive afferents innervating the glabrous skin of the hand can be categorized into one of four types: fast-adapting type 1 (FA1), fast-adapting type 2 (FA2), slowly-adapting type 1 (SA1), and slowly-adapting type 2 (SA2) (*Vallbo and Johansson, 1984*). In intraneural microstimulation (INMS), single mechanoreceptive afferents are selectively activated by passing a small (1–7 µA) current through the recording microelectrode, thus evoking a quantal sensation in the projected sensory field, which matches the physiological qualities of the recorded mechanoreceptive afferent (*Torebjörk et al., 1987*). Microstimulation of an FA1 afferent evokes a well-defined, local sensation of 'flutter' or 'buzzing', while microstimulation of an SA1 afferent evokes a sensation of continuous pressure or inward pulling (*Vallbo et al., 1984*; *Ochoa and Torebjörk, 1983*). Microstimulation of an FA2 afferent evokes a diffuse sensation of vibration over a larger area, whereas microstimulation of an SA2 afferent does not produce a consistent, conscious sensory experience (*Vallbo et al., 1984*; *Ochoa and Torebjörk, 1983*).

It has been shown in a small number of previous studies that INMS of single mechanoreceptive afferents can be combined with noninvasive imaging methods to advance our understanding of the effects of mechanoreceptive afferent activity in somatosensory cortices. For example, INMS of FA1 and SA1 afferents in the median nerve produces frequency-following electroencephalography responses within contralateral S1 (*Kelly et al., 1997*). The single previous study combining INMS with fMRI (*Trulsson et al., 2001*), using a 3 T scanner and a surface coil positioned over the parietal lobe contralateral to the site of stimulation, showed that INMS of FA1 and SA1 afferents induced activity in S1 and S2, which overlapped with regions activated by applying mechanical vibration to the relevant units' receptive fields. However, a detailed characterization of the specificity of single unit INMS activations within the representation of the digits in S1 has yet to be performed.

Several studies have previously assessed the cortical response to vibrotactile stimulation of the glabrous skin of the human hand, and shown that this evokes a hemodynamic response in multiple primary and secondary cortical areas, including contralateral S1, bilateral S2, primary motor cortex (M1), supplementary motor area (SMA), cingulate cortex, posterior parietal cortex (PPC), and insula cortex (*McGlone et al., 2002*; *Trulsson et al., 2001*; *Gelnar et al., 1998*). Ultra-high field (7T) fMRI has also recently been used in conjunction with vibrotactile stimulation to map individual digit representations and resolve the fine, within-digit organization (base-to-tip), thus revealing functional subdivisions of areas in S1 (*Sanchez-Panchuelo et al., 2010*; *Sanchez-Panchuelo et al., 2012*).

Compared to lower field measurements, 7T fMRI provides greatly increased sensitivity and blood-oxygenation level dependent (BOLD) signal contrast, coupled with improved intrinsic spatial specificity (*Gati et al., 1997*). Here, we used 7T fMRI to resolve whole-brain cortical activation patterns evoked by INMS of single mechanoreceptive afferent units in the glabrous skin of the hand, and to assess the precise spatial localization of INMS-evoked BOLD responses in contralateral S1, in comparison to activation due to mechanical vibrotactile stimulation.

## Results

A total of 33 mechanoreceptive afferents were found (17 FA1, 14 SA1, 1 FA2 and 1 SA2) in 4 participants during 10 experimental sessions. We focused our study on the cortical response to stimulation of type 1 afferents (FA1 and SA1), as these units are far more numerous in the volar hand than type 2 units (FA2 and SA2) (*Vallbo and Johansson, 1984*). Example recordings from FA1 and SA1 units are shown in *Figures 1a and b* respectively, demonstrating that good quality signals can be recorded from single mechanoreceptive afferents in the environment of a 7T magnetic resonance scanner. INMS of single units produced distinct sensations: FA1 stimulation was typically felt as vibration or buzzing, while SA1 stimulation elicited a sensation of pressure or pulling (see *Table 1*).

Due to the technically challenging set-up (e.g. 2 units were lost on moving the participant into the scanner bore) and the nature of the method (e.g. the stimulated unit corresponds to the unit from which recordings were previously made only around 50% of the time [*Torebjörk et al., 1987*]), INMS was carried out during concurrent fMRI in 11 units (U1-U11) that gave single-point sensations,

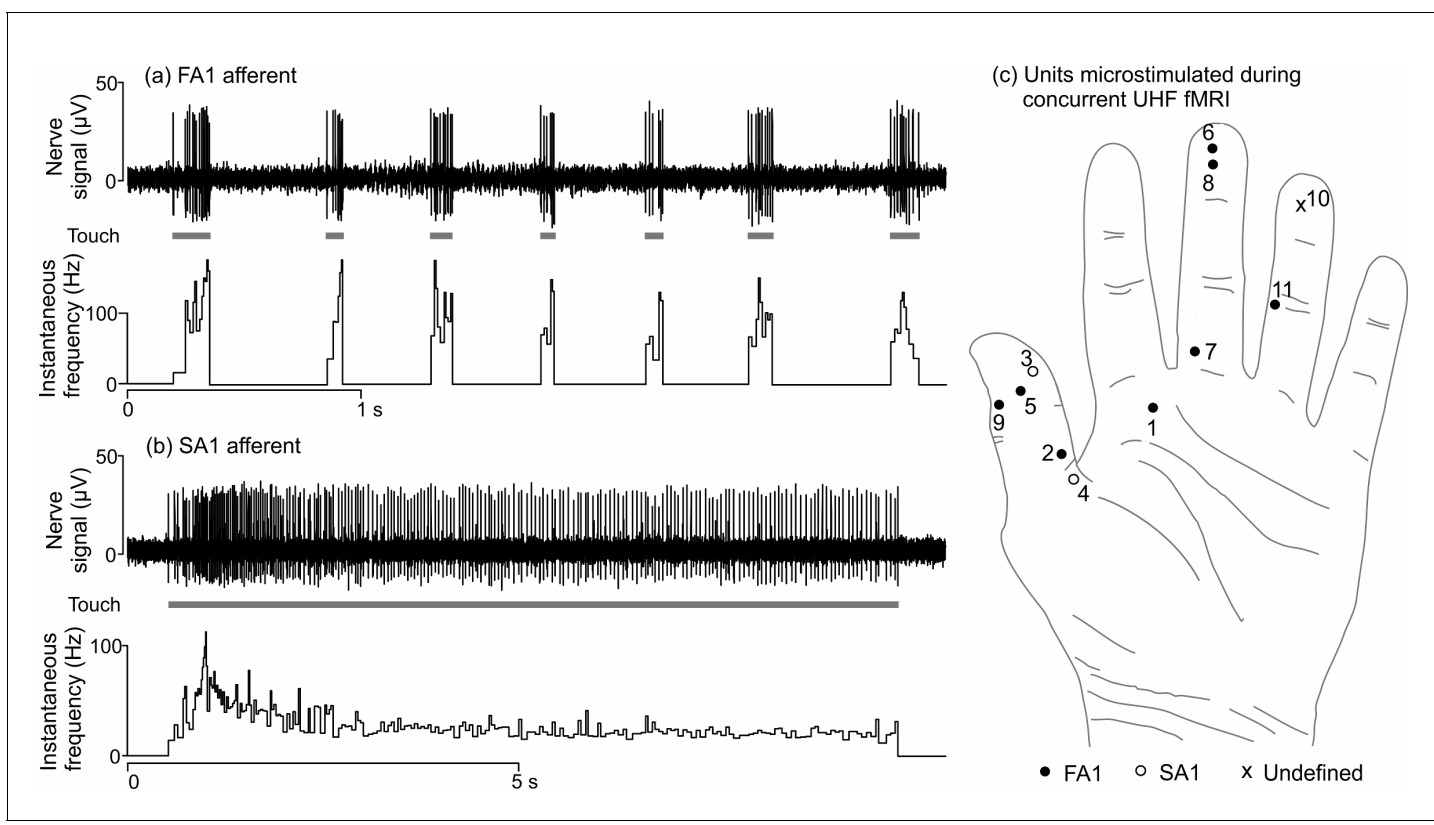

**Figure 1.** Physiological recordings from mechanoreceptive afferents and the location of afferents that were microstimulated during 7T fMRI. Example microneurography recording (top) along with the instantaneous firing frequency (bottom) for (**a**) an FA1 afferent (U1; see *Table 1*) and (**b**) an SA1 afferent collected inside the 7T MR scanner environment. In (**a**), mechanical taps were delivered to the center of the FA1's receptive field and (**b**) a long-lasting mechanical indentation was applied at the center of the SA1's receptive field, using a wooden stick (see gray blocks). (**c**) Location of the afferents that were microstimulated during 7T fMRI (see *Table 1*). U9 was located on the right hand, but has been transposed onto the left hand for this schematic. The 'undefined' (x) afferent relates to a sensation that was felt as a line, which likely indicates two single afferents in close proximity being stimulated simultaneously.

**Table 1.** Mechanoreceptive afferent units in which INMS was performed during 7T fMRI. The table details the unit type and location, as well as the frequency and perception of applied INMS. All units were located on the left hand unless stated.

| Participant | Unit | Type | Location | Physiology | Sensation | Frequency |
|---|---|---|---|---|---|---|
| 1 | 1 | FA1 | Palm | Yes | Buzzing | 30 Hz |
| 2 | 2 | FA1 | Base of digit 1 | Yes | Small dots | 60 Hz |
| | 3 | SA1 | Middle of digit 1 | Yes | Pulling | 30 Hz |
| | 4 | SA1 | Base of digit 1 | Yes | Pulling | 30 Hz 60 Hz |
| | 5 | FA1 | Middle of digit 1 | Yes | Vibration | 60 Hz |
| | 6 | FA1 | Digit 3 fingertip | No | Tapping, vibration | 30 Hz 60 Hz 90 Hz |
| 3 | 7 | FA1 | Base of digit 3 | Yes | Small, round point of tingle sensation | 30 Hz |
| | 8 | FA1 | Digit 3 fingertip | No | Small, round point of tingle sensation | 30 Hz 60 Hz 90 Hz |
| 4 | 9 | FA1 | Middle of digit 1 (*right hand*) | No | Prickle, flutter | 30 Hz |
| | 10 | Undefined | Digit 4 fingertip | No | Small line* | 30 Hz |
| | 11 | FA1 | Middle of digit 4 | No | Flutter | 30 Hz |

*A small line sensation is indicative of the simultaneous stimulation of two afferents that are in close proximity.

6 of which were electrophysiologically-characterized (see *Table 1*). The receptive field locations for these units are shown in *Figure 1c*.

## Cortical responses to single unit INMS and vibrotactile stimulation in S1

Clear and reproducible BOLD responses were found in somatosensory regions, when INMS was perceived. Occasionally, participants reported that the sensation evoked by the INMS stopped, likely due to a minor dislodgement of the microelectrode. This occurred for U7 where a projected sensation was perceived prior to scanning, but no sensation was felt during the fMRI run. For some units, the sensation was weak (U2, U3; possibly due to difficulty in attending to the stimulus sensation when inside the scanner), or lost during the fMRI run (U5, U6, U8). We compared the location of fMRI responses of all perceived INMS units in contralateral S1 with the digit representation obtained from both vibrotactile stimulation of the microstimulated unit's receptive field and the fMRI somatotopy maps formed from the traveling-wave (phase-encoding) vibrotactile paradigm (*Figure 2*). We found that fMRI responses to INMS of single units (all except for U1; *Figure 3—figure supplement 1*) were spatially localized within the relevant S1 digit representation identified from vibrotactile stimulation. *Figure 2a* shows example maps of digit somatotopy defined from the vibrotactile traveling-wave paradigm for Participant 4 in the right and left hemispheres (left and right of the figure, respectively). *Figure 2b* shows the BOLD response to INMS of U11 (right) and U9 (left) for Participant 4. These responses are well-localized within regions of the somatotopic map for digit 4 of the left hand and digit 1 of the right hand, respectively. *Figure 2c* shows the activation generated in S1 by applying vibrotactile stimulation to the receptive field of U11 (right) and U9 (left). Fits to the hemodynamic responses evoked in S1 by INMS and the application of vibrotactile stimulation to the unit's receptive field can be seen in *Figure 2d*.

*Figure 3* shows the spatial localization of the activation produced in S1 by the seven perceived INMS units (U4-U6, U8-U11) (*Figure 3a*) and corresponding vibrotactile stimulation of each units' receptive field (*Figure 3b*). In general, the BOLD responses due to INMS and vibrotactile stimulation were well localized within the expected digit ROI, as defined from the traveling-wave somatotopy paradigm. *Figure 3c* plots the average INMS z-score (FDR corrected) in each digit ROI, and *Figure 3d* shows the proportion of active voxels to the INMS paradigm that were classified to each digit ROI (z>3.08, FDR corrected). As expected, the average z-score and proportion of active voxels

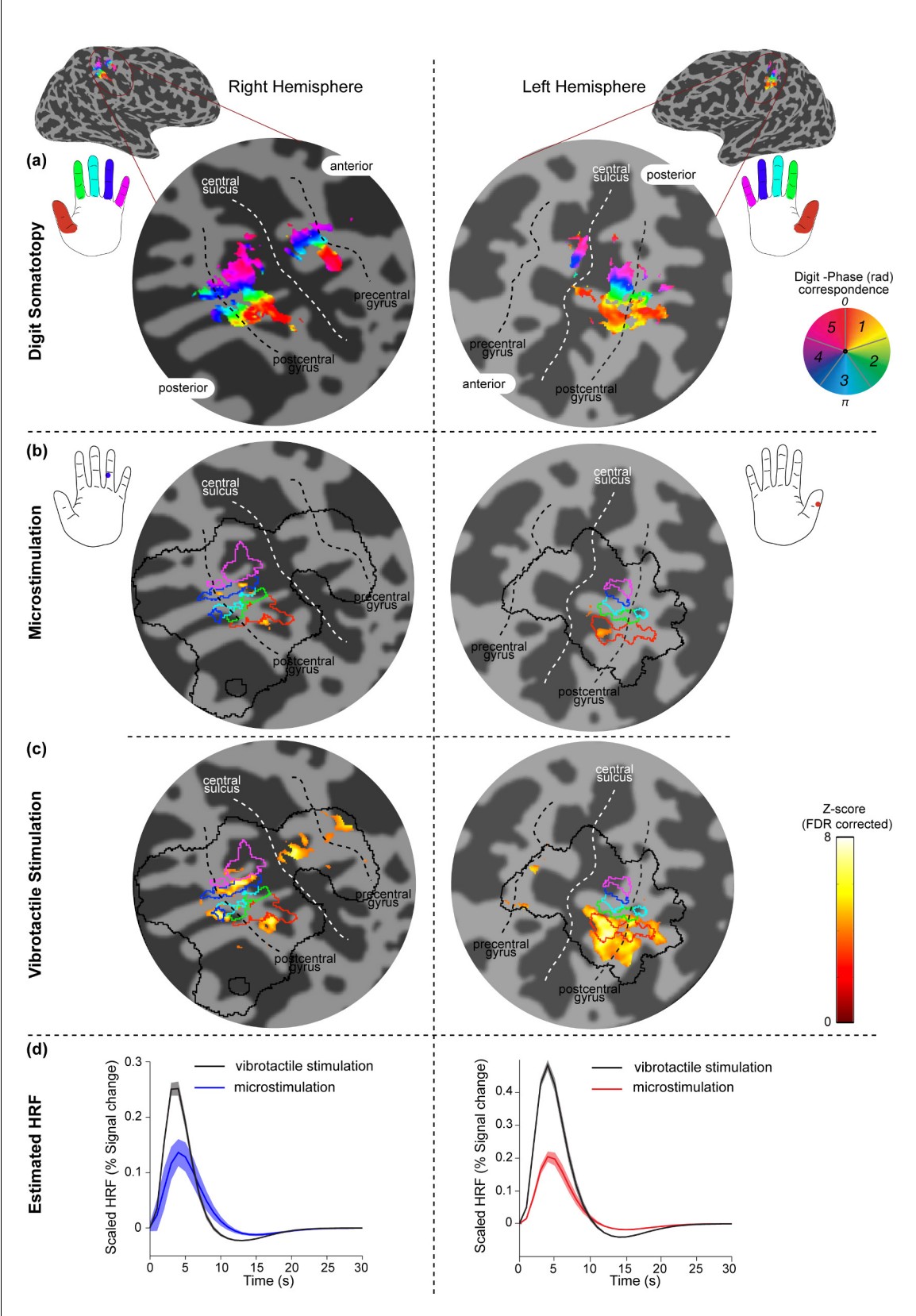

**Figure 2.** Spatial localization of INMS-induced versus vibrotactile-induced responses in contralateral S1. Activation maps related to stimulation of two different afferents in Participant 4 are rendered onto a flattened cortical patch spanning the central sulcus of the right (left of figure) and left (right of *Figure 2 continued on next page*

*Figure 2 continued*

figure) hemispheres. Dark gray represents the sulci and light gray the gyri. (a) Digit somatotopy, where phase values (in radians) and corresponding preferred stimulus location (fingertip) are shown. Orderly representation of the digits is found on the posterior bank of the central sulcus (white line) and the post-central gyrus (dashed black line), corresponding to S1. (b) Statistical maps (Z > 3.08, FDR-adjusted) from INMS of U11 (left) and U9 (right). BOLD activation is localized within the expected digit ROI identified from digit somatotopy, as shown by the blue (digit 4) and red (digit 1) lines, which denote phase values encoded by the blue (3.77–5.03 rad) and orange (0–1.26 rad) colors respectively. The solid black line indicates the SI hand mask (calculated by dilating the somatotopy map by 5 voxels) within which FDR correction was performed. (c) Statistical maps (Z > 3.08, FDR-adjusted) for vibrotactile stimulation of the corresponding receptive fields of U11 (top) and U9 (bottom). (d) HRF estimated from the GLM analysis for INMS and vibrotactile stimulation averaged across voxels of the ROI (U10, top; U9, bottom). Error bars show voxel-wise parameter standard errors averaged across voxels of the ROI.

in the digit ROIs corresponding to digits in which the INMS was sensed was higher than in the neighboring digits. *Figure 4* plots the group-level response to show the spatial spread of the INMS and vibrotactile response to neighboring digits. *Figure 4a* shows the mean z-score, *Figure 4b* the proportion of active voxels and *Figure 4c* the GLM parameter estimate to INMS (top) and vibrotactile stimulation of the unit's receptive field (bottom). ANOVA results showed a significant difference in mean Z-score ($F_{4,30}$=14.08, p<$10^{-5}$; $F_{4,30}$=12.97, p<$10^{-5}$), proportion of active voxels ($F_{4,30}$=16.12, p<$10^{-6}$; $F_{4,30}$=17.64, p<$10^{-6}$) and GLM parameter estimates ($F_{4,30}$=13.52, p<$10^{-5}$; $F_{4,30}$=14.1, p<$10^{-5}$) across the stimulated and neighboring digit classification (INMS; vibrotactile). A multiple pairwise comparison, adjusted for multiple comparisons, showed that measures for the stimulated digit were significantly higher than those of the neighboring digits for mean Z-score (p<0.0001 INMS; p<0.005 vibrotactile stimulation), proportion of active voxels (p<0.00005 for INMS and vibrotactile stimulation) and GLM parameter estimates (p0.01 for INMS and vibrotactile stimulation).

For those units lost during the fMRI run (U5, U6, U8), no areas were found to show a significant correlation with an additional (parametric) regressor when modelling linear reductions in induced response over time (to model gradual losses of unit responses), likely due to the sudden rather than gradual loss of the unit. Thus parameter estimates to INMS stimulation were not significantly different between the GLM including a parametric regressor and modelling the INMS stimulation alone.

## Comparison of cortical activity patterns between single unit INMS and vibrotactile stimulation

Participants freely described the mechanical, point-vibrotactile stimulus applied to each unit's receptive field as feeling very similar in extent and quality to the INMS, especially for the sensations generated from FA1 units. *Figure 5a* compares the mapping of INMS-induced fMRI responses (yellow) for all FA1 single units to maps of the responses produced by applying vibrotactile stimulation to the units' receptive fields (blue). Overlapping cortical responses are shown in green. Activation maps show the conjunction of the individual FA1 unit responses, using the same statistical threshold (Z > 3.08, false discovery rate (FDR) correction) for both INMS and vibrotactile stimulation. BOLD responses to single unit INMS were detected in a number of sensory-related brain areas, including S1, S2 (Brodmann areas (BA) 40 and 43), premotor cortex (PMC; SMA and dorsal PMC), M1, insula (anterior insula cortex (AIC) and posterior insula cortex (PIC)), prefrontal cortex (PFC) and PPC. *Table 2* details the location and statistical significance (mean and standard error across units) of the BOLD responses produced in these areas by INMS of the five FA1 single units in the left hand. Common areas of activation for INMS and vibrotactile stimulation included S1, S2, PMC, M1, and contralateral PIC; however, INMS gave rise to significant activity in additional brain regions, including the AIC, PPC and contralateral PFC (*Table 2*). *Figure 5b* shows that the HRFs generated in these regions by INMS were similar in both onset and duration to the INMS-elicited responses in S1 and S2.

## Discussion

The principal finding of our present work is the detailed localization in contralateral S1 of cortical responses to the electrical microstimulation of single, first-order mechanoreceptive afferents, and the demonstration of spatial alignment of these responses with somatotopic maps derived from mechanical skin stimulation. This was achieved through the combined usage of two techniques:

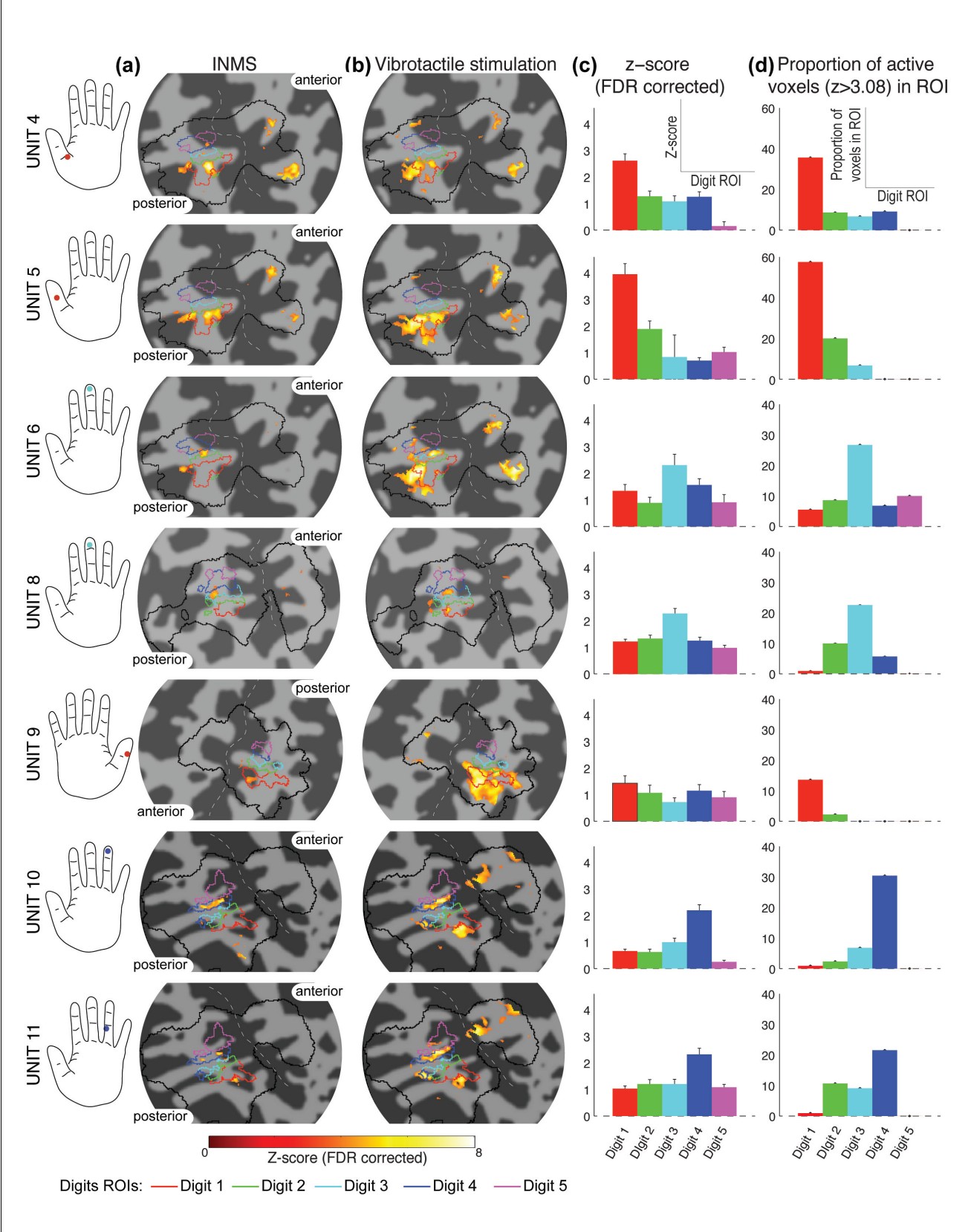

**Figure 3.** Spread of activation across the digit ROIs identified from the somatotopy. (**a**) Statistical maps (Z > 3.08, FDR-adjusted) from INMS of seven single units in participants 2, 3 and 4. In each case the activation map is rendered onto a flattened cortical patch spanning the central sulcus of the right
*Figure 3 continued on next page*

*Figure 3 continued*
hemisphere. Dark gray represents the sulci and light gray the gyri. The solid black line indicates the SI hand mask (calculated by dilating the somatotopy map by 5 voxels) within which FDR correction was performed. Activation is localized within the expected digit ROI (black line) identified from the digit somatotopy (see color legend). (**b**) Statistical maps (Z > 3.08, FDR-adjusted) for vibrotactile stimulation of the corresponding receptive field of units. (c) Z-scores (FDR-corrected) of the INMS BOLD response averaged across voxels for each of the digit ROIs identified from the traveling-wave analysis. Error bars indicate standard error across voxels in ROI. (**d**) Proportion of voxels activated by the INMS paradigm at Z>3.08 (FDR-corrected) for each digit ROI. The source data for plots in panels (c) and (d) are available in the ***Figure 3—source data 1***.
The following source data and figure supplement are available for figure 3:

**Source data 1.** Source files for plots of Z-score and Proportion of active voxels in each Digit ROI.
**Figure supplement 1.** Comparison of contralateral S1 responses to different paradigms for Participant 1.

intra-neural microstimulation (INMS), to stimulate single mechanoreceptive afferents, and 7T fMRI, to map the cortex with superior spatial resolution. This work also shows that activity generated by stimulation of a single mechanoreceptive afferent can be perceptually characterized and produces a network of cortical responses.

Only one previous study has combined single unit INMS with fMRI, at 3T (***Trulsson et al., 2001***), but this was only able to resolve activation in contralateral S1 and S2 as the use of a surface coil limited the spatial extent of activation maps. The greater signal-to-noise ratio and improved BOLD contrast afforded by 7T fMRI allowed us to improve the spatial resolution, with a reduction in the voxel volume by a factor of 6 compared to previous work at 3T (***Trulsson et al., 2001***). We have exploited the improved spatial resolution to provide a detailed characterization of the location and extent of the cortical network involved in encoding inputs from single mechanoreceptive afferents, as well as in comparing these responses to somatotopical maps created from vibrotactile skin stimulation.

Measurements of cortical activity elicited by INMS demonstrated that when a singular, quantal touch from the stimulation of a single mechanoreceptive afferent is consciously felt, a precise area in contralateral S1 is active. The response in S1 was well-localized within the expected region, identified from maps of digit somatotopy obtained from vibrotactile stimulation of the fingertips. The extent of the S1 responses to INMS was less than that elicited by vibrotactile stimulation to the unit's receptive field, although the response produced by single unit INMS was relatively extensive, considering that vibrotactile stimulation simultaneously engages a large number of afferents (***Johansson and Vallbo, 1979***; ***Vallbo and Johansson, 1984***).

Robust responses were found within the expected digital cortical area for all perceived microstimulated afferents (***Figures 2*** and ***3***), except for U1, for which no significant responses were found, in either contralateral or ipsilateral S1, despite the fact that the participant exhibited a complete somatotopic map of the digits in both hemispheres and reported feeling the sensation throughout INMS. To explore this finding further, we used the delineation of digit 2 from the somatotopic map obtained with the vibrotactile traveling-wave paradigm to inspect the time series of S1 responses evoked by INMS for U1 (located on the palm below digit 2). We also interrogated the BOLD response produced in contralateral S1 when vibrotactile stimulation was applied to the receptive fields of U1. In S1, we found negative BOLD responses (***Figure 3—figure supplement 1***) for both INMS and vibrotactile stimulation applied to the receptive field of the INMS. The negative BOLD response in this subject is possibly due to a steal effect from the nearby vasculature draining from the active cortex (***Bianciardi et al., 2011***) since draining venous regions are highly modulated by block paradigms with periods of 'on' and 'off' stimulation, as used to study the response to INMS and vibrotactile stimulation of the receptive field. In contrast, using the traveling-wave paradigm a complete map of the digits in S1 is seen. This is expected, as we have previously shown that a traveling-wave design is insensitive to the non-specific BOLD contributions from large veins that drain blood from across the whole hand representation in S1 (***Uğurbil et al., 2003***; ***Besle et al., 2013***), thus suppressing the venous signal modulations found in the block INMS/vibrotactile stimulation data. In order to estimate the spatial spread of INMS BOLD responses to neighboring digits, we show that, at the group level, the z-score, proportion of active voxels and GLM parameter estimates are significantly higher (p<0.01) in the stimulated ROI than in the neighboring digits (***Figure 4***).

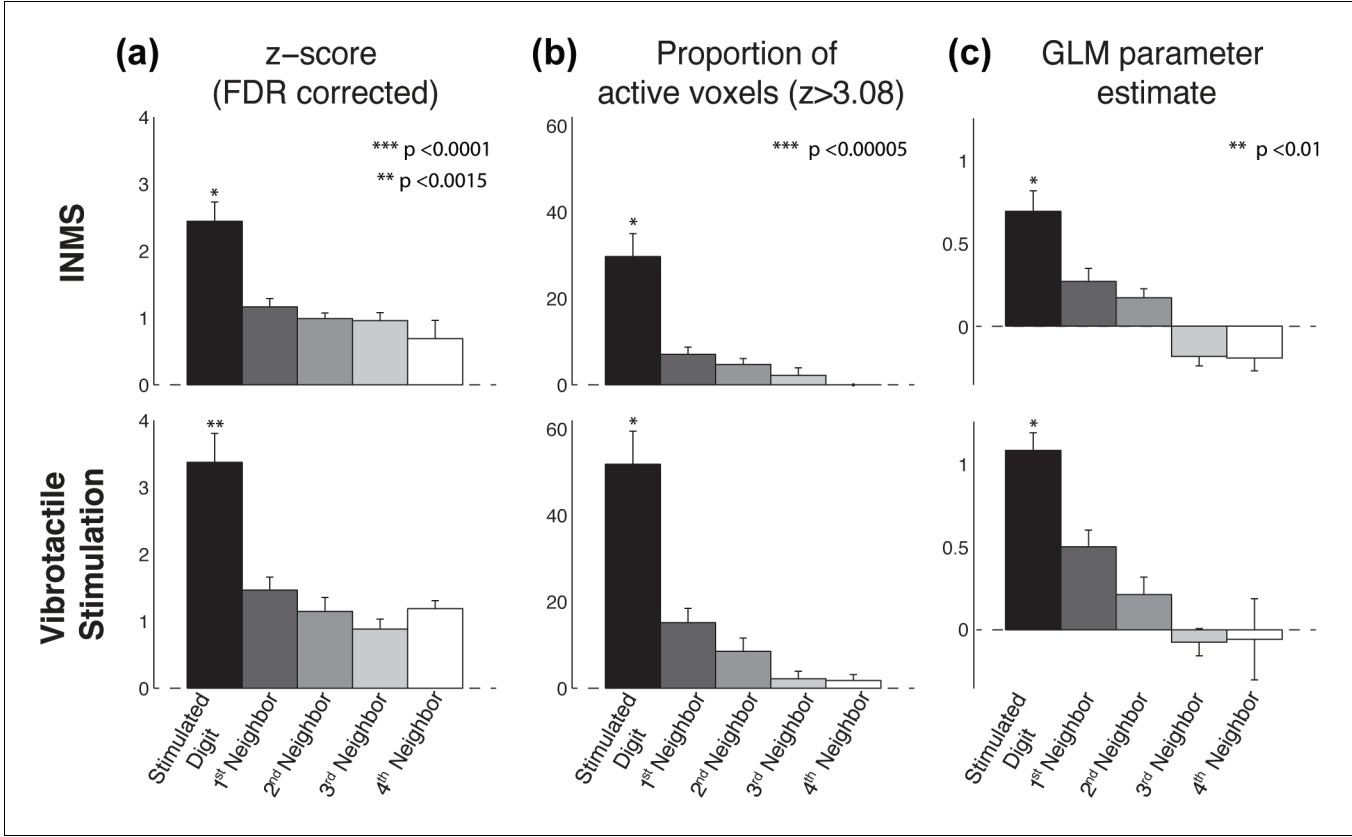

**Figure 4.** Group analysis (N = 7 units) of the BOLD response to INMS and vibrotactile stimulation of the unit's receptive field, showing the stimulated digit compared to the neighboring digits. (a) Z-scores (FDR-corrected) of INMS response in digit ROIs (defined from digit somatotopy) averaged across ROIs for the stimulated digit (N = 7) compared to neighboring digits (1st degree neighbors, N = 11; 2nd degree neighbors, N = 9, 3rd degree neighbors, N = 5, 4th degree neighbors, N = 3). The z-score for the stimulated digit was significantly different to that of neighboring digits. ***p<0.0001, **p<0.005, statistical significance corrected for multiple comparison using Bonferroni correction. (b) Proportion of voxels activated by the INMS (top) and vibrotactile (bottom) paradigm at Z>3.08 (FDR-corrected) for the stimulated digit compared to the neighboring digits. Mean and standard error across ROIs. The proportion of active voxels in the stimulated digit ROI was significantly different to that of neighboring digits. ***p<0.00005, statistical significance corrected for multiple comparison using Bonferroni procedure. (c) GLM parameter estimates of the INMS (top) and vibrotactile (bottom) paradigm for the stimulated digit compared to the neighboring digits. The parameter estimate in the stimulated digit ROI was significantly higher than that of neighboring digits. **p<0.01, statistical significance corrected for multiple comparison using Bonferroni procedure. For all plots (a–c) the mean and standard error across N measures is shown. The source data used for the ANOVA tests are available in the *Figure 4—source data 1*.

The following source data is available for figure 4:

**Source data 1.** Source files for ANOVA tests.

These results are in-line with our previous findings reported for vibrotactile stimulation (*Besle et al., 2013*).

The network of cortical areas activated by both INMS of single mechanoreceptive afferents and mechanical vibrotactile stimulation of the units' receptive field, included somatosensory areas such as S1, S2, and PIC, as well as areas involved in motor control, including M1, SMA and PMC. Although M1 has previously been shown to be activated by tactile input (e.g. *Francis et al., 2000*; *Ackerley et al., 2012*), we cannot exclude the possibility that the M1 activation observed in this study may originate from spatial blurring of somatosensory activation (given that M1 and S1 are located on opposite banks of the central sulcus). When comparing responses to INMS and vibrotactile stimulation applied to the afferents' receptive fields, INMS activated a number of additional areas, specifically the AIC, PPC and PFC. Exploration of the INMS BOLD time series for these areas (*Figure 5b*) suggests that the activity in these areas is locked to the S1/S2 activity and is not due to

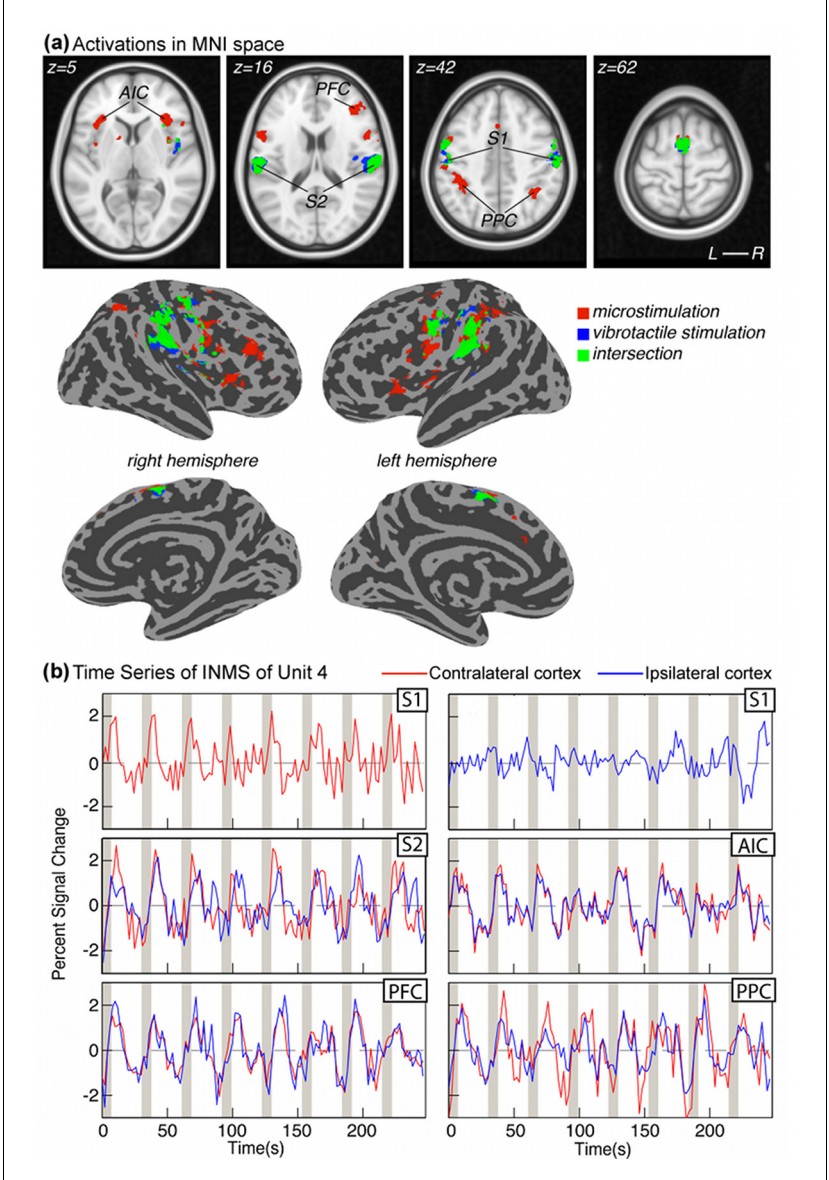

**Figure 5.** fMRI activation patterns and time courses in cortical areas. (a) Cortical activation patterns in MNI space. Transverse slices and surface reconstructions showing areas of activation in response to INMS (red clusters) and mechanical vibrotactile stimulation applied directly to the respective unit's receptive field (blue clusters), as well as areas of overlap (green clusters). Clusters represent common regions of significant activation from all single FA1 units on the left hand (U1, U4, U6, U8, and U11). Individual statistical maps for each afferent were thresholded at Z < 3.08 after correcting for multiple comparisons (FDR) and cluster-corrected at p=0.01, prior to forming the conjunction map. (b) BOLD time courses due to INMS for U4 in different cortical areas. Responses contralateral (right) to the hand stimulation site are shown in red and ipsilateral (left) responses are shown in blue.

anticipation. Both insula and parietal cortices have been shown to contribute to the perception of touch (*Preusser et al., 2014*), and a previous study of tactile attention (*Burton et al., 2008*) has shown that a fronto-parietal network, which includes PFC and PPC, is involved in attention. Although identical paradigm timings were used for INMS and vibrotactile stimulation in order to compare the spatial localization of the BOLD response, there were differences in the attentional focus between the INMS and vibrotactile tasks. During the INMS fMRI runs, participants were aware that perception might be lost and hence had to concentrate on the stimulus and report any lack of sensation at the

**Table 2.** Cortical areas showing significant activation to INMS of single mechanoreceptive afferents and the corresponding vibrotactile stimulation. Results show the mean and standard error across the five FA1 mechanoreceptive afferents subject to INMS at 30 Hz and corresponding vibrotactile stimulation of the perceived sensation, showing the number of units showing significant activation, MNI coordinates, beta values, Z-score and number of voxels in ROI. R=contralateral, L=ipsilateral. Source files for **Table 2—source data 1** and **Table 2—source data 2** contain single unit INMS and vibrotactile stimulation results, respectively, for each of the 5 (U1, U4, U6, U8, U11) individual units.

| ROI | No. Units | x, y, z MNI co-ordinates | Single unit INMS | | | Vibrotactile stimulation | | |
|---|---|---|---|---|---|---|---|---|
| | | | Beta | Z | Voxels | Beta | Z | Voxels |
| SI R | 4 | 54, -12, 46 | 1.4 ± 0.2 | 5.9 ± 0.5 | 38 ± 7 | 1.3 ± 0.3 | 5.4 ± 0.3 | 41 ± 12 |
| SI L | 3 | -52, -12, 44 | 1.2 ± 0.2 | 5.6 ± 0.8 | 20 ± 9 | 1.6 ± 0.3 | 5.2 ± 0.2 | 19 ± 1 |
| BA 40 R | 5 | 60, -22, 16 | 1.4 ± 0.2 | 4.9 ± 0.2 | 56 ± 5 | 1.4 ± 0.1 | 4.8 ± 0.2 | 54 ± 7 |
| BA 40 L | 4 | -60, -22, 16 | 1.5 ± 0.4 | 5.3 ± 0.2 | 73 ± 5 | 1.4 ± 0.2 | 5.0 ± 0.1 | 72 ± 12 |
| BA 43 R | 2 | 60, -4, 10 | 1.1 ± 0.4 | 5.4 ± 0.1 | 45 ± 6 | 1.2 ± 0.4 | 4.4 ± 0.2 | 30 ± 20 |
| BA 43 L | 3 | -58, -12, 14 | 1.0 ± 0.4 | 4.8 ± 0.3 | 33 ± 8 | 1.7 ± 0.3 | 4.2 ± 0.2 | 26 ± 11 |
| SMA R | 5 | 4, 0, 60 | 1.2 ± 0.2 | 4.8 ± 0.3 | 93 ± 27 | 1.3 ± 0.2 | 4.8 ± 0.2 | 43 ± 21 |
| SMA L | 5 | -2, 0, 60 | 1.2 ± 0.2 | 4.5 ± 0.3 | 66 ± 19 | 1.2 ± 0.1 | 4.5 ± 0.3 | 29 ± 6 |
| PMC R | 4 | 54, 0, 50 | 0.8 ± 0.2 | 4.7 ± 0.2 | 36 ± 11 | 1.1 ± 0.2 | 5.0 ± 0.2 | 46 ± 9 |
| PMC L | 5 | -52, -2, 50 | 1.1 ± 0.1 | 5.5 ± 0.3 | 37 ± 7 | 1.2 ± 0.1 | 4.3 ± 0.1 | 20 ± 8 |
| M1 R | 3 | 54, -6, 48 | 0.9 ± 0.2 | 5.2 ± 0.5 | 51 ± 20 | 0.8 ± 0.2 | 5.0 ± 0.7 | 31 ± 10 |
| M1 L | 2 | -52, -6, 48 | 1.5 ± 0.2 | 6.3 ± 0.1 | 66 ± 36 | 1.3 ± 0.1 | 5.3 ± 0.5 | 21 ± 3 |
| PIC R | 5 | 46, -2, 10 | 0.8 ± 0.2 | 4.2 ± 0.2 | 45 ± 12 | 0.8 ± 0.2 | 4.7 ± 0.2 | 27 ± 3 |
| PIC L | 5 | -42, -2, 10 | 0.8 ± 0.1 | 4.4 ± 0.2 | 38 ± 14 | - | - | - |
| AIC R | 4 | 34, 26, 4 | 1.2 ± 0.1 | 4.7 ± 0.2 | 146 ± 20 | - | - | - |
| AIC L | 4 | -32, 26, 4 | 1.1 ± 0.1 | 4.4 ± 0.2 | 106 ± 21 | - | - | - |
| PPC R | 4 | 38, -48, 50 | 1.2 ± 0.1 | 4.4 ± 0.3 | 168 ± 44 | - | - | - |
| PPC L | 5 | -38, -48, 56 | 1.0 ± 0.1 | 4.4 ± 0.3 | 172 ± 43 | - | - | - |
| PFC R | 4 | 42, 34, 18 | 1.2 ± 0.2 | 4.5 ± 0.3 | 78 ± 22 | - | - | - |

Source data 1. Source files for single unit INMS. This matlab file contains 2D-matrices (19x5) with the results for single unit INMS for each of the 5 individual units (U1, U4, U6, U8, U11) in each of the 19 ROIs. 'BetaValues' contains mean across voxels of the beta values, 'Z-score' contains the mean Z_score (FDR- corrected) across voxels and 'NumberVoxels' contains the number of significant active voxels (Z > 3.08, FDR-corrected) in the ROI. **Table 2** summarizes the results by showing the mean and standard error across the 5 units.

Source data 2. Source files for vibrotactile stimulation. This matlab file contains 2D-matrices (19 ROIs x 5 units) with the results for vibrotactile stimulation applied to the receptive field for each of the 5 individual units (U1, U4, U6, U8, U11) in each ROI. 'BetaValues 'contains mean across voxels of the beta values, 'Z_score' contains the mean Z-score (FDR- corrected) across voxels and 'NumberVoxels' contains the number of significant active voxels (Z > 3.08, FDR-corrected) in the ROI. **Table 2** summarizes the results by showing the mean and standard error across the 5 units.

end of the run. In contrast, the vibrotactile stimulus was delivered at a suprathreshold level and participants did not have to monitor that the sensation was still present during the vibrotactile fMRI run. Hence, the increased activity in AIC, PFC and PPC observed in the present study may reflect the increased attentional effects (i.e., baseline or gain effects on evoked responses) during the INMS protocol compared to vibrotactile stimulation. However, this is a preliminary finding and requires further investigation with larger sample sizes and more quantitative analysis to be corroborated.

The capability of combining INMS with 7T fMRI has the following theoretical implications for human somatosensory research. Although the notion that peripheral input from the skin is represented directly by four cytoarchitectonic areas (BA 3a, 3b, 1 and 2) in S1, each containing an orderly somatotopic map of the body surface has been supported by findings from animal studies (**Kaas et al., 1979**; **Paul et al., 1972**; **Favorov et al., 1987**; **Tommerdahl et al., 2010**) and 7T fMRI in humans (**Sanchez-Panchuelo et al., 2010**; **Sanchez-Panchuelo et al., 2012**; **Martuzzi et al., 2014**), a simple point-to-point topographical correspondence between skin surface and cortical

representation does not hold. In reality, there is integration and processing through axonal synapsing in the dorsal column nuclei and thalamus prior to mechanoreceptive information entering the cerebral cortex. There appears to be a preserved transmission from single, mechanoreceptive second-order neurons in the dorsal column (*Vickery et al., 1994*). At the level of the thalamus, an axon of a single ventral posterolateral nucleus terminates over a fairly wide, roughly 0.5 mm, cortical territory (*Rausell and Jones, 1995*), where many individual thalamocortical axons spread out in discrete patches over several millimeters of S1 (*Landry et al., 1987*). This spread corresponds well with our finding that the cortical activation from a single mechanoreceptive afferent extends over an area that is not dissimilar to the area activated by input from many afferents through point-vibrotactile stimulation. Also, neurons in S1 cortical columns have extensive lateral excitatory connections, not only with neighboring neurons, but also with neurons several millimeters away in the same cortical area (*Burton and Fabri, 1995*). We have shown that single unit INMS produces bilateral somatosensory activation, as well as influencing motor areas and cognitive networks (e.g. PPC, PFC). Such a wide spreading of stimulus-evoked activity has been clearly documented in microelectrode recording studies (*Reed et al., 2010*). Overall, the spatiotemporal pattern of S1 response to vibrotactile stimulation is far from simple and its functional significance remains to be unraveled.

Translational insights from in vivo neurophysiological studies in non-human primates have driven much of the theoretical understanding of cortical mechanisms that govern human tactile perception, but operative procedures, especially those which alter the neurochemistry of cortical synaptic transmission (*Masamoto et al., 2009*), may confound relating such findings to normal functioning of the human brain. This demonstration of the feasibility of combining INMS with 7T fMRI opens up the possibility of a range of further neuroimaging studies that will allow interrogation of the precise anatomical and physiological properties of the fundamental encoding of touch. These include systematic investigation of the sub-cortical (e.g. thalamic) responses and laminar-specific cortical responses to INMS of different mechanoreceptive afferent classes using a variety of electrical stimulation patterns.

## Materials and methods

Ten experimental sessions were conducted on four right-handed participants (30–64 years, 2 male). Procedures were approved by the University of Nottingham Medical School Ethics Committee and all participants gave full, written, informed consent. Due to the precision needed in performing INMS within the magnetic resonance scanner, participants were required to lie extremely still and feel relaxed; all participants were accustomed to the fMRI environment (two participants had participated in INMS experiments previously). Each experimental session involved three steps: (1) microneurography for the characterization of a single mechanoreceptive afferent (*Vallbo and Hagbarth, 1968*); (2) assessment of the sensation to INMS; (3) concurrent INMS and fMRI. Participants subsequently took part in a second fMRI session in which vibrotactile stimulation was delivered.

Participants lay on the scanner bed with their arm (the left arm in all cases except one experiment on the right arm) immobilized using cushions. Survey, reference and $B_0$-map scans were acquired, and an image-based shimming approach (*Sanchez-Panchuelo et al., 2010*) used to minimize magnetic field inhomogeneity, with the optimized shim currents remaining fixed throughout the subsequent fMRI runs. The participant was moved out of the bore of the magnet to perform Steps (1) and (2).

### Microneurography

In Step 1, the median nerve was accessed at the wrist in order to isolate single axonal responses from mechanoreceptive afferents in the volar hand, on which to perform INMS (*Trulsson et al., 2001*). A high-impedance (~300–500 kΩ), insulated, tungsten recording/stimulating electrode (15 mm length, shaft diameter 0.2 mm, tip diameter ~5 μm; FHC, Bowdoin, ME) was inserted percutaneously into the skin, ~3 cm from the wrist fold between the flexor carpi radialis and the flexor palmaris longus tendons. An uninsulated reference electrode was inserted subcutaneously 3–5 cm away, on the ulnar side of the recording/stimulating electrode, and a ground electrode was attached further up the participant's arm (*Figure 6*). The recording/stimulating electrode was advanced into the median nerve, which was located 0.3–1 cm below the skin surface. The preamplifier was taped to the participant's arm, and the acquisition hardware and stimulator were located at the outer edge of the scanner room (*Figure 6*). Differential responses were amplified (x10,000) using a preamplifier

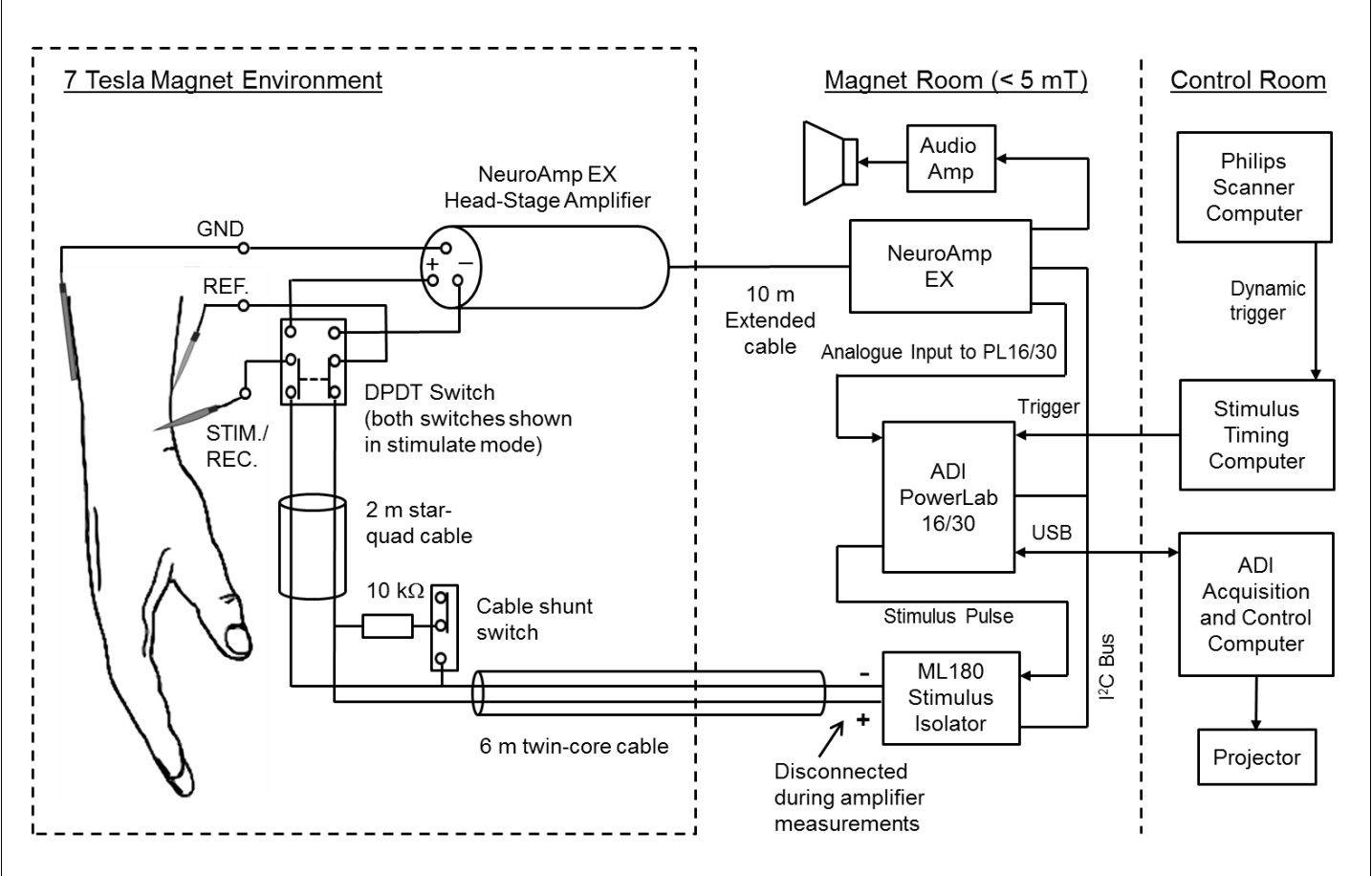

**Figure 6.** Figure of the experimental setup. The PowerLab, NeuroAmp EX and ML180 stimulator were placed just inside the magnet room at a field strength not exceeding 5 mT. Placement of the interface equipment within the magnet room was preferred for safety reasons, as isolated cables connected to the participant did not then pass into the control room. The USB interface and trigger cables were passed through the radio frequency shield via a waveguide aperture. An amplifier and loudspeaker was driven from the NeuroAmp EX audio output to give audio feedback to the microneurographer. In addition, a projection of the computer screen could be viewed for visual confirmation of nerve signals. A switch was used to connect the electrodes to either the stimulator or the NeuroAmp head-stage pre-amplifier. In addition, a resistive shunt was placed across the stimulation leads to remove any build-up of charge before connecting or disconnecting the stimulator. Disconnection of the stimulator was necessary because of the high level of noise introduced when it was connected. Star-quad cable was used within the magnet environment to reduce the likelihood of induced currents due to scanner operation affecting the stimulus presentation.

(NeuroAmpEX; ADInstruments, Castle Hill, Australia), band-pass filtered (0.3–5 kHz) and sampled at 10 kHz using PowerLab hardware and LabChart 7 software (ADInstruments, Castle Hill, Australia).

The microneurographer delivered light, stroking touch to the palm to evoke activity in low-threshold mechanoreceptive afferents. A loudspeaker in the scanner room allowed the microneurographer to hear the nerve activity and a projector displayed the recording onto the scanner exterior for visual inspection. The microneurographer systematically searched for the nerve until modulations of the signal from the electrode corresponded to mass activity from mechanoreceptive afferents as a result of touch were heard. Using fine adjustments, the electrode was manipulated within the nerve to an intra-fascicular location and single units were searched for by stroking the participant's hand.

Single mechanoreceptive afferents were characterized by their audio and visual signals, and the extent of the receptive field of each afferent was explored using a wooden stick. The location of the receptive field was mapped using von Frey monofilaments and the minimal force required for mechanoreceptor activation noted. Afferents were identified as being myelinated Aβ mechanoreceptors, namely FA1, SA1, FA2 or SA2 afferents (**Vallbo and Johansson, 1984**). The middle of the receptive field was marked on the skin. Recordings of individual mechanoreceptive afferents in response to

mechanical stimulation were made (e.g. *Figure 1a,b*) and analyzed in MATLAB (The Mathworks; Natick, MA). Data were preprocessed to verify the single-unit nature of all recorded mechanoreceptive afferents with an offline pattern-matching algorithm.

## Single unit INMS

Once a single mechanoreceptive afferent was identified, INMS was carried out to ascertain the sensation produced by a low-current electrical pulse sequence (Step 2). Trains of 30 Hz pulses (200 μs, positive, square-wave pulses over 0.5 s) were delivered (via Stimulus Isolator; ADInstruments, Castle Hill, Australia and controlled using the LabChart 7 software). The experimenter delivered 2–3 pulse sequences, while the current was increased slowly from 0 μA, in 1 μA steps, until the participant felt a sensation. Once a clear sensation was felt, the precise location of the sensation and its quality were recorded and tested to confirm whether the previously mapped receptive field spatially aligned with that perceived by the participant during INMS. This was done by a process of questioning the participant to determine whether mechanical touch to the receptive field matched the projected sensory field sensation during INMS to within ~1 mm. If so, it was deemed that microstimulation was being applied to the afferent from which recordings had been made. If the participant felt a clear small, point-sensation in the projected sensory field that did not align with the mapped receptive field, the stimulated unit was nevertheless explored. These units were included if the perceived sensation (e.g. pressure from an SA1) was similar in quality to those in matched physiology-INMS trials (e.g. perceived size, shape, sensation) (see *Table 1*). The stimulating current intensity which generated a sensation was recorded, along with the stimulation currents delivered during each fMRI run. INMS of a stable, single mechanoreceptive afferent could be carried out successfully for up to ~45 min, although Step 3 was completed successfully for only a subset of mechanoreceptive afferents (see Results).

## fMRI paradigm

Each fMRI run consisted of a block paradigm, comprising 8 cycles of alternating periods of 8 s INMS followed by 23 s rest (acquisition time ~4 mins). The 8 s INMS period consisted of 0.5 s burst of stimulation (30 Hz pulse frequency; 200 μs pulse width) each second. For each afferent, 1–3 fMRI repeats of the INMS paradigm were conducted. In some cases, the stimulation current was adjusted between runs, e.g. due to loss of perception (*Vallbo et al., 1984*), to ensure a clear and stable sensation. If the INMS-induced sensation remained stable, other parameters were also tested, including changing the stimulation frequency to 60 Hz, and increasing the stimulation current to investigate the effect of recruiting further mechanoreceptive afferents (*Vallbo et al., 1984*).

After Steps 1–3, fMRI of mechanical vibrotactile stimulation at each microstimulated afferent's receptive field was carried out with identical timings to the INMS paradigm. Vibrotactile stimuli were delivered at 30 Hz to ~1 mm$^2$ of the skin using a piezo-electric device (Dancer Design, St-Helens, UK). In addition, the digit tips of each participant's left hand (and right hand for participant 4) were stimulated with 5 independently-controlled piezo-electric devices using a traveling-wave or phase-encoding paradigm (*Sanchez-Panchuelo et al., 2010*), analogous to that used in retinotopic mapping, in which each individual digit of the hand is sequentially stimulated to create a travelling wave of activity across cortical regions containing a somatotopic map of the hand. Vibrotactile stimulation at 30 Hz was delivered to each digit tip in periods of 4 s (intermittent stimulation with 0. 1 s gap every 0.5 s), over a 20 s cycle. Data were collected during two runs of 12 cycles each; with stimulation delivered in a forward (digit 1 to 5) and reverse order (digit 5 to 1).

## fMRI acquisition

MRI data were collected on a 7T scanner (Achieva; Philips, Amsterdam, Netherlands) using a head volume transmit coil and 32-channel receive coil (Nova Medical; Wilmington, MA). Functional data were acquired using $T_2^*$-weighted, multi-slice, single-shot gradient-echo, echo-planar imaging (EPI) with echo time (TE) 25 ms, repetition time (TR) 2000 ms, flip angle (FA) 75°, SENSE reduction factor 3 in the right-left direction. The in-plane spatial resolution was 1.5 mm, field of view of 174 × 192 mm$^2$ in right-left and anterior-posterior directions. A slice thickness of 2.5 mm was used to achieve full brain coverage (80 mm in foot-head direction) within the TR period. For the traveling-wave

paradigm, the slice thickness was reduced to 1.5 mm (48 mm coverage) as it was only necessary to span S1.

Following the functional runs, a high-resolution $T_2^*$-weighted FLASH dataset was acquired with the same slice prescription and coverage as the functional data ($0.5 \times 0.5 \times 1.5$ mm$^3$ resolution; TE/TR = 9.3/458 ms, FA = 32°, SENSE factor = 2), and a whole-head structural $T_1$-weighted MPRAGE dataset (1 mm isotropic resolution, linear phase encoding order, TE/TR 3.7/15 ms, FA 8°, inversion time 1184 ms, TR-FOCI pulse [*Hurley et al., 2010*]) to allow projections of functional maps onto flattened reconstructions of the cortical space and MNI space.

fMRI raw time series and structural MRI scans for each subject can be found at figshare (Sanchez Panchuelo, RM; Ackerley, R; Glover, PM; Bowtell, RW; Wessberg, J; Francis, ST; McGlone, F | 2016 | fMRI to intraneural microstimulation of single mechanoreceptive afferents | Available at figshare under a CC0 Public Domain.)

## fMRI data analysis

fMRI data sets were realigned to the last volume of the data set using AFNI (http://afni.nimh.nih.gov/afni), and statistical analysis performed using mrTools (http://www.cns.nyu.edu/heegerlab) in MATLAB. To account for scanner drift and other low-frequency signals, all time-series were high-pass filtered (0.01 Hz cut-off) and data converted to percent signal change. To address the key aims, three analyses were performed:

### Cortical responses to single unit INMS and vibrotactile stimulation in S1

The spatial localization of microstimulated afferents in S1 was compared with digit somatotopic maps formed for each participant using a traveling-wave paradigm (*Sanchez-Panchuelo et al., 2010*). The somatotopic map was used to define ROIs specific to each of the 5 digits of the hand, these were subsequently used as independent ROIs to allow group-level inference tests to be conducted (as performed in *Besle, 2013*). Here, data were not spatially smoothed in order to retain high spatial resolution. Both the INMS data, and data acquired during vibrotactile stimulation applied to the skin location where each afferent was perceived, were analyzed using a general linear model (GLM) employing a canonical HRF model and its orthogonalized temporal derivative. FDR adjustment (*Benjamini and Hochberg 1995*) was performed using an adaptive step-up method (*Benjamini et al., 2006*). All adjusted P-values were converted to quantiles of standard normal distribution (Z-score). Analysis was restricted to voxels identified using the traveling-wave localizer (dilated by 5 voxels to ensure complete coverage of the S1 hand area) to reduce the number of inference tests on both the INMS and vibrotactile stimulation data to compute FDR corrected Z-scores. We investigated the spread of INMS induced activations, and vibrotactile stimulation to each unit's receptive field, by computing the mean Z-score, proportion of active voxels, and GLM parameter estimates in each digit ROI. Subsequently, to quantify spread of responses into neighboring digits at the group-level, INMS and vibrotactile responses for the ROI corresponding to the stimulated digit were combined, by averaging the mean Z-score, proportion of active voxels, and GLM parameter estimates (N=7 units; 3 Digit 1 ROIs, 2 Digit 3 ROIs, 2 Digit 4 ROIs). This procedure was then repeated for the 1$^{st}$ degree (N = 11), 2$^{nd}$ degree (N = 9), 3$^{rd}$ degree (N = 5) and 4$^{th}$ degree (N = 3) neighboring digit ROIs. A one-way analysis of variance (ANOVA) tests was then performed on this data, and post-hoc multiple pairwise comparison, adjusted for multiple comparisons using Bonferroni correction.

For those units for which the stimulus sensation was lost during the fMRI run, a further GLM analysis was run which included a regressor of linear parametric modulation in time, and the associated parameter estimates were assessed.

Functional statistical maps from each microstimulated afferent and the traveling-wave localizer were rendered onto flattened representations of the central sulcus obtained using the mrFlatMesh algorithm (VISTA software, http://white.stanford.edu/software/) based on cortical segmentations from the whole head $T_1$-weighted anatomical data obtained using Freesurfer (http://surfer.nmr.mgh.harvard.edu/). Having aligned functional data to the participant's whole head $T_1$-weighted anatomical reference volume (see *Alignment of functional data*), statistical maps were transformed to flattened space using linear interpolation and displayed at the central cortical depth.

## Whole brain analysis

This was performed to compare those brain areas responding to INMS of a single mechanoreceptive afferent with those responding to vibrotactile stimulation. Data were spatially smoothed with a Gaussian FWHM 3 mm and a second GLM analysis was performed on the whole volume for both the INMS data and the vibrotactile stimulation data to the unit's receptive fields. The resulting Z-score statistical maps were threshold at Z<3.08 after FDR-adjustment and cluster-correction (p<0.01) to visualize activation maps and to compute binary masks for each stimulated mechanoreceptive unit (and for corresponding vibrotactile stimulation to each unit's receptive field).

Functional statistical maps from all five single FA1 afferents of the left hand stimulated during INMS at 30 Hz (U1, U4, U6, U8, and U11) were projected onto standard MNI space to identify active brain areas from probabilistic brain atlases (Harvard-Oxford cortical structure and Talairach Daemon labels, in FSL). Functional maps were transformed into the participant's whole head anatomical reference volume (see *Alignment of functional data*). The whole-head anatomical $T_1$-weighted MPRAGE from each participant was aligned to a standard $T_1$-weighted MNI template using first an affine FLIRT registration, followed by a FNIRT non-linear registration algorithm (FSL, FNIRT). This alignment was then applied to the statistical maps from the participant's INMS unit to warp the data into standard MNI space. A map was computed of the intersection of responses to all five FA1 afferents, from which to define significant regions of interest (ROIs). These ROIs were transformed to native EPI space for each individual afferent and the beta values, Z-scores and number of active voxels were interrogated for each significant ROI, in each afferent's native space. Similarly, the corresponding BOLD maps resulting from vibrotactile stimulation applied to the skin location where each afferent was perceived were transformed into MNI space and identical analyses performed.

## Alignment of functional data to participant's whole head anatomical reference volume

Statistical maps were moved from functional acquisition space into whole-head anatomical $T_1$-weighted space for detailed comparison with digit somatotopy in flattened reconstructions of the cortical space and for combination in standard MNI space (see *Whole brain analysis*). We estimated the alignment between the (distorted) reference EPI volume from the motion correction and the undistorted $T_2^*$-weighted anatomical volume using FNIRT. Functional maps were non-linearly transformed into structural $T_2^*$-weighted volume space using FNIRT's 'applywarp' and then linearly transformed from the structural $T_2^*$-weighted to whole-head $T_1$-weighted volume space. Note that this registration was only used for the display of statistical maps; all statistical analyses of functional data were performed in native EPI space.

## Acknowledgements

Funding: Pain Relief Foundation grant, Medical Research Council, Vetenskapsrådet (Swedish Research Council), Royal Society International Exchanges scheme. Gratitude is extended to Professor Oleg Favorov for invaluable discussions on the manuscript.

## Additional information

### Funding

| Funder | Grant reference number | Author |
|---|---|---|
| Medical Research Council | Programme Grant, G9900259 | Paul M Glover Richard W Bowtell Susan T Francis |
| Pain Relief Foundation | R254046 | Francis McGlone |
| Royal Society | International Exchange, IE130419 | Rosa Maria Sanchez Panchuelo Rochelle Ackerley |
| Vetenskapsrådet | K2013-62X-03548 | Johan Wessberg |

| VGR jointly with European Comission | MoRe, FP7/2007-2013 under REA grant agreement n. 608743 | Rochelle Ackerley |

The funders had no role in study design, data collection and interpretation, or the decision to submit the work for publication.

## Author contributions
RMSP, Contributed to the design of the study, Developed the magnetic resonance sequences and acquired the MRI data, Performed all the fMRI/MRI data analysis, Contributed to the interpretation of the results and drafted the paper; RA, Contributed to the design of the study, Conducted the microneurography and analyzed the microneurography data, Contributed to the interpretation of the results and drafting of the paper; PMG, Contributed to the design of the study and collection of the data, Adapted the standard microneurography equipment for compatibility with an ultra-high field MRI scanner, Contributed to the interpretation of the results and revised the paper; RWB, Contributed to the design of the study and collection of the data, Contributed to the interpretation of the results and revised the paper; JW, Contributed to the design of the study and collection of the data, Conducted the microneurography, Contributed to the interpretation of the results and revised the paper; STF, Contributed to the design of the study, Developed the magnetic resonance sequences and acquired the MRI data, Guided the fMRI analysis and contributed to the interpretation of the results and writing of the paper; FM, Contributed to the design of the study and collection of the data, Contributed to the interpretation of the results and writing of the paper

## Author ORCIDs
Rosa Maria Sanchez Panchuelo, http://orcid.org/0000-0002-9917-0020
Rochelle Ackerley, http://orcid.org/0000-0003-4621-7929

## Ethics
Human subjects: This work was approved by the University of Nottingham Medical School Ethics Committee. All participants gave full, written, informed consent.

# Additional files

## Major datasets
The following dataset was generated:

| Author(s) | Year | Dataset title | Dataset URL | Database, license, and accessibility information |
|-----------|------|---------------|-------------|--------------------------------------------------|
| Sanchez Panchuelo RM, Ackerley R, Glover PM, Bowtell RW, Wessberg J, Francis ST, McGlone F | 2016 | fMRI to intraneural microstimulation of single mechanoreceptive afferents | https://dx.doi.org/10.6084/m9.figshare.3205552 | Available at figshare under a CC0 Public Domain licence |

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
