## [Decision Letter]

Thank you for submitting your work entitled "Mapping quantal touch using 7 Tesla functional magnetic resonance imaging and single-unit intraneural microstimulation" for consideration by *eLife*. Your article has been favorably evaluated by Jody Culham (Senior editor) and three reviewers, one of whom, Eilon Vaadia, is a member of our Board of Reviewing Editors.

The following individual involved in review of your submission has agreed to reveal their identity: Karl Friston (peer reviewer).

The reviewers have discussed the reviews with one another and the Reviewing Editor has drafted this decision to help you prepare a revised submission.

Summary:

The authors demonstrate a system that provides high resolution of monitoring cortical activity in response to localized activation of single mechanoreceptors in humans. The combination of 7-T field imaging, INMS and tactile stimulation provides a very impressive methodological advance, by demonstrating the feasibility of microneurography and fMRI and validating the approach against conventional tactile stimulation. The report establishes a nice correspondence between responses to microstimulation in S1, S2 and equivalent afferent input produced by somatosensory (vibrotactile) simulation. The nice thing about this correspondence is that the afferent input is carefully characterized using microneurography prior to scanning. The microneurography and MRI physics in this study are superb.

However, as noted at the initial submission stage, given methodological strengths and limited theoretical advances, the paper was viewed as more appropriate for the "Tools and Resources" format at *eLife*, which should provide clear and transparent descriptions of the technological advance and demonstrate that it works. This view was further reinforced by the external reviewers, who also thought that the paper's theoretical advances were too limited for a Research Article format.

The comments below address the required revisions of the manuscript as a "Tools and Resources" article. Most importantly, the strongest advance of the present work over past work from your group is the validation of the INMS technique compared to mechanical stimulation; however, at present, these comparisons are largely qualitative and need more robust statistical validation. The extension to 7-T scanning and the recruitment of a broader range of brain areas are interesting, but in consideration of potential confounds that could drive these areas, the arguments need some tempering.

Essential revisions:

1) Data Analysis

A) SI topographic specificity: Further characterization of the specificity of the INMS activation within the digit maps is lacking and at present the manuscript relies too heavily on example data (e.g., Figure 2) that, while compelling, do not convey how reliable the effects were. How consistent was the correspondence between the digit map finger-selectivity and the INMS activity across units? For example, did all units show topographically appropriate activity in SI? How much did the activity "leak" to other finger representations? Statistical quantification can improve the validity of the major findings. Given that the authors are arguing that the somatosensory representation of the two stimuli (INMS and vibrotactile) was similar, a direct comparison between the two modalities, and a contrast between two different locations within each modality seems more suitable to support this point. That said, as the stimulation used to construct the digit maps was on the fingertips, and the receptive fields (RFs) of the units were often more medial, I'm not sure the observed overlap is sensible. This warrants further consideration.

B) Although the results are sufficient to provide a description of the induced BOLD responses – and the correspondence between microstimulation and vibrotactile stimulation, the authors could have finessed the analysis in a number of ways. For example, if subjects reported the perceptual intensity changed during the fMRI scanning, it would have been better to perform a parametric analysis; testing for the effects of perception on induced responses. In other words, instead of simply showing that the activations in a particular subject reduced when they ceased to feel the microstimulation, you could have tested the hypothesis that perception was a key determinant of BOLD responses by using it as a parametric regressor. Similarly, when comparing the microstimulation with vibrotactile responses, instead of simply saying they are similar (or showing areas of conjunction) you could have modeled both in a single model. This would have allowed you to report the differences between microstimulation and somatosensory stimulation explicitly – by looking at the interaction between two experimental factors (on versus off and microstimulation versus vibrotactile stimulation). If you feel comfortable with the analyses above, it would be simple to say; for example, "the differences between microstimulation and vibrotactile stimulation were tested formally in a general linear model of both time series (testing for interaction) and confirmed the recruitment of extra areas during microstimulation".

C) There are concerns about the meaning of the intriguing results showing activation in numerous higher order cortical areas, and the potential misinterpretation of the physiological mechanisms by which the brain interpret inputs, even from single mechanoreceptors. Given the activation of motor cortical areas, the response to INMS may depend on other factors (e.g., action and adaptive prediction, attention, arousal) by which the system may modulate the perceived response. The experiments and the data described here are not sufficient to address these points. Although one could continue this study further adding experiments where the subject has to respond by a movement or vice versa, (initiate a movement in order to trigger the INMS), the journal discourages requests for new experiments that are not essential to the paper. Therefore, the reviewers suggest leaving these findings for future experiments. You could briefly mention the finding and clearly state that these are preliminary results, suggesting that the method can be used in future experiments for quantitative studies.

Suggested revisions:

Given the limited data you have, the analyses of the currents you used and the frequency of microstimulation may be better left until another study (where you vary the intensity and frequency orthogonally in a factorial design). The three reviewers recommend to keep the focus on the stronger points (as summarized by the authors in the first section of the Discussion, and as we, the reviewers summarize above).

[Editors' note: further revisions were requested prior to acceptance, as described below.]

Thank you for resubmitting your work entitled "Mapping quantal touch using 7 Tesla functional magnetic resonance imaging and single-unit intraneural microstimulation" for further consideration at *eLife*. Your revised article has been favorably evaluated by Jody Culham (Senior editor), Eilon Vaadia (Reviewing editor), and two reviewers. The following individuals involved in review of your submission have agreed to reveal their identity: Karl Friston (Reviewer #2); Tamar R Makin (Reviewer #3).

This is an improved version of a previous submission. The authors appear to have dealt with many of the reviewers concerns. The new Figure 3 and Figure 4 in particular provide a highly compelling proof of concept. Yet, there are a couple of outstanding points that could be clarified. We suggest that the authors respond to these recommendations, and resubmit a final revised version.

Comment 1:

At our request, the authors have included an additional analysis but it is introduced before being described in a confusing way (subsection “Cortical responses to single unit INMS and vibrotactile stimulation in S1”, third paragraph. For example, you could say:

"no areas were found to show a significant correlation with an additional (parametric) regressor modelling linear reductions in induced responses over time (to model gradual losses of unit responses)."

Comment 2:

The authors need to describe what they mean by "A travelling wave design".

To prevent confusion with the spatial and temporal modelling of the sort used in phase encoding and retinotopic mapping, we would recommend a footnote or insertion of the following sort:

"A travelling wave design simply involves short blocks of (four seconds) of ordered sequential stimulation to each digit, which is modelled in the usual way using boxcar stimulus functions."

Comment 3:

In the third paragraph of the Introduction the authors describe the potential impact of their study given previous research. They highlight the necessity for detailed study of cortical activity beyond SI due to INMS. While this might be true, this is not the main strength of the current study. We therefore suggest that instead, they highlight the necessity of detailed investigation in the SI digit maps.

---

## [Author Response]

1) Data Analysis

A) SI topographic specificity: Further characterization of the specificity of the INMS activation within the digit maps is lacking and at present the manuscript relies too heavily on example data (e.g., Figure 2) that, while compelling, do not convey how reliable the effects were. How consistent was the correspondence between the digit map finger-selectivity and the INMS activity across units? For example, did all units show topographically appropriate activity in SI? How much did the activity "leak" to other finger representations? Statistical quantification can improve the validity of the major findings. Given that the authors are arguing that the somatosensory representation of the two stimuli (INMS and vibrotactile) was similar, a direct comparison between the two modalities, and a contrast between two different locations within each modality seems more suitable to support this point. That said, as the stimulation used to construct the digit maps was on the fingertips, and the receptive fields (RFs) of the units were often more medial, I'm not sure the observed overlap is sensible. This warrants further consideration.

The correspondence between the digit selectivity and the INMS response is consistent across units, except for U1 as described in the Discussion.

In Figure 3, we now include all 7 units for which INMS was perceived, showing the spatial correspondence between activation due to INMS and vibrotactile stimulation of each unit’s receptive field, as well as the individual digit ROIs derived from the traveling-wave paradigm. We have previously shown the spatial overlap of the representations of individual digits in S1 using vibrotactile stimulation (Besle et al., ‘Single-subject fMRI mapping at 7 T of the representation of fingertips in S1: a comparison of event-related and phase-encoding designs’, J Neurophys, 2013; Besle et al., ‘Event-Related fMRI at 7T Reveals Overlapping Cortical Representations for Adjacent Fingertips in S1 of Individual Subjects’, HBM, 2013). Applying similar analysis methods in this manuscript, we now use the independent ROIs derived from the traveling-wave paradigm to investigate the spread of activation to neighbouring digit ROIs during INMS and vibrotactile stimulation. In Figure 3) and (D) we now plot quantitative results of the average FDR-corrected z-score and the proportion of active voxels in each digit ROI. In Figure 4, we combine the data from all units at the group level to show that the z-score, proportion of active voxels and GLM parameter estimates are significantly higher in the ROI of the stimulated digit than in the ROIs of the neighboring digits. This is in-line with the findings we previously reported for vibrotactile stimulation.

We have also previously shown that the extent of activation in response to vibrotactile stimulation of the phalanges of the index finger (digit 2) from tip-to-base is encompassed within the digit 2 ROI obtained from a traveling-wave paradigm applied to the finger tips (Figure 2 (Sanchez Panchuelo et al., ‘Within-digit functional parcellation of Brodmann areas of the human primary somatosensory cortex using fMRI at 7 Tesla’, J Neurosci, 2012)). This is because S1 is divided into 4 Brodmann areas (3a/3b/1/2), with the tip of the finger being overrepresented in comparison to the base. It is thus appropriate to use ROIs derived from fingertip stimulation in a traveling-wave paradigm in order to assess the spatial localisation of more medial INMS units.

B) Although the results are sufficient to provide a description of the induced BOLD responses – and the correspondence between microstimulation and vibrotactile stimulation, the authors could have finessed the analysis in a number of ways. For example, if subjects reported the perceptual intensity changed during the fMRI scanning, it would have been better to perform a parametric analysis; testing for the effects of perception on induced responses. In other words, instead of simply showing that the activations in a particular subject reduced when they ceased to feel the microstimulation, you could have tested the hypothesis that perception was a key determinant of BOLD responses by using it as a parametric regressor. Similarly, when comparing the microstimulation with vibrotactile responses, instead of simply saying they are similar (or showing areas of conjunction) you could have modeled both in a single model. This would have allowed you to report the differences between microstimulation and somatosensory stimulation explicitly – by looking at the interaction between two experimental factors (on versus off and microstimulation versus vibrotactile stimulation). If you feel comfortable with the analyses above, it would be simple to say; for example, "the differences between microstimulation and vibrotactile stimulation were tested formally in a general linear model of both time series (testing for interaction) and confirmed the recruitment of extra areas during microstimulation".

We have now performed a parametric analysis, including a regressor of linear modulation in time, on the three datasets in which subjects’ perception was lost during the fMRI run (U5, U6 and U8). However, no areas were found to display a significant suppression, and so parameter estimates were similar (within error) for the GLM, with and without this regressor. This is most likely to be because the loss of perception was not gradual, but rather perception was completely lost within a single cycle. We have now described this additional GLM analysis in the Methods and added a sentence to the Results to indicate that this analysis did not reveal a significant effect for U5, U6, and U8.

We have not modelled the microstimulation and vibrotacile data in a single GLM, as the microstimulation responses are variable in amplitude across units, adding between-subject variance to a INMS group map, and are also of lower amplitude than for vibrotactile stimulation. Thus we believe that the analysis we have carried out is more appropriate in this case for the high resolution INMS and vibrotactile stimulation data.

C) There are concerns about the meaning of the intriguing results showing activation in numerous higher order cortical areas, and the potential misinterpretation of the physiological mechanisms by which the brain interpret inputs, even from single mechanoreceptors. Given the activation of motor cortical areas, the response to INMS may depend on other factors (e.g., action and adaptive prediction, attention, arousal) by which the system may modulate the perceived response. The experiments and the data described here are not sufficient to address these points. Although one could continue this study further adding experiments where the subject has to respond by a movement or vice versa, (initiate a movement in order to trigger the INMS), the journal discourages requests for new experiments that are not essential to the paper. Therefore, the reviewers suggest leaving these findings for future experiments. You could briefly mention the finding and clearly state that these are preliminary results, suggesting that the method can be used in future experiments for quantitative studies.

We have considerably reduced this paragraph and have added a statement in the manuscript to reflect this point: ‘However, this is a preliminary finding and requires further investigation with larger sample sizes and more quantitative analysis to be corroborated.’

Suggested revisions:

Given the limited data you have, the analyses of the currents you used and the frequency of microstimulation may be better left until another study (where you vary the intensity and frequency orthogonally in a factorial design). The three reviewers recommend to keep the focus on the stronger points (as summarized by the authors in the first section of the Discussion, and as we, the reviewers summarize above).

As suggested by the reviewers, we now focus more on the characterization of the specificity of the INMS activation to the vibrotactile digit maps (results added to Figure 3 and Figure 4), and have removed the previous Figure 5 (effect of increasing the stimulation current) and Figure 3—figure supplement 2 (effect of increasing frequency).

[Editors' note: further revisions were requested prior to acceptance, as described below.]

Comment 1:

At our request, the authors have included an additional analysis but it is introduced before being described in a confusing way (subsection “Cortical responses to single unit INMS and vibrotactile stimulation in S1”, third paragraph. For example, you could say:

"no areas were found to show a significant correlation with an additional (parametric) regressor modelling linear reductions in induced responses over time (to model gradual losses of unit responses)."

We have replaced the relevant sentence in the manuscript with that suggested by the reviewers.

Comment 2:

The authors need to describe what they mean by "A travelling wave design".

To prevent confusion with the spatial and temporal modelling of the sort used in phase encoding and retinotopic mapping, we would recommend a footnote or insertion of the following sort:

"A travelling wave design simply involves short blocks of (four seconds) of ordered sequential stimulation to each digit, which is modelled in the usual way using boxcar stimulus functions."

The travelling wave design is in fact equivalent to the phase encoding design used for retinotopic mapping. To clarify this, we have added ‘(phase encoding)’ next to the first mention of the travelling wave design (subsection “Cortical responses to single unit INMS and vibrotactile stimulation in S1”, first paragraph) and the paragraph that describes the traveling-wave design in the Materials and methods section now reads: ‘…. a traveling-wave or phase-encoding paradigm (Sanchez-Panchuelo et al. 2010), analogous to that used in retinotopic mapping, in which each individual digit of the hand is sequentially stimulated to create a travelling wave of activity across cortical regions containing a somatotopic map of the hand’.

Comment 3:

In the third paragraph of the Introduction the authors describe the potential impact of their study given previous research. They highlight the necessity for detailed study of cortical activity beyond SI due to INMS. While this might be true, this is not the main strength of the current study. We therefore suggest that instead, they highlight the necessity of detailed investigation in the SI digit maps.

As suggested we have changed this section to highlight the necessity of a detailed characterization of the specificity of single unit INMS activations within the representation of the digits in S1.